# Highly Efficient Cardiac Differentiation and Maintenance by Thrombin-Coagulated Fibrin Hydrogels Enriched with Decellularized Porcine Heart Extracellular Matrix

**DOI:** 10.3390/ijms24032842

**Published:** 2023-02-02

**Authors:** Fatemeh Navaee, Philippe Renaud, Alexander Kleger, Thomas Braschler

**Affiliations:** 1Microsystems Laboratory-LMIS4, EPFL, 1015 Lausanne, Switzerland; 2Department of Pathology and Immunology, Faculty of Medicine, CMU, 1211 Geneva, Switzerland; 3Institute of Molecular Oncology and Stem Cell Biology, Ulm University Hospital, 89081 Ulm, Germany; 4Interdisciplinary Pancreatology, Department of Internal Medicine 1, Ulm University Hospital, 89081 Ulm, Germany; 5Organoid Core Facility, Medical Faculty, Ulm University Hospital, 89081 Ulm, Germany

**Keywords:** dECM-fibrin hydrogel, neonatal cardiomyocyte culture, H9c2 cell differentiation, beating synchrony, 3D co-culture

## Abstract

Biochemical and biophysical properties instruct cardiac tissue morphogenesis. Here, we are reporting on a blend of cardiac decellularized extracellular matrix (dECM) from porcine ventricular tissue and fibrinogen that is suitable for investigations employing an in vitro 3D cardiac cell culture model. Rapid and specific coagulation with thrombin facilitates the gentle inclusion of cells while avoiding sedimentation during formation of the dECM-fibrin composite. Our investigations revealed enhanced cardiogenic differentiation in the H9c2 myoblast cells when using the system in a co-culture with Nor-10 fibroblasts. Further enhancement of differentiation efficiency was achieved by 3D embedding of rat neonatal cardiomyocytes in the 3D system. Calcium imaging and analysis of beating motion both indicate that the dECM-fibrin composite significantly enhances recovery, frequency, synchrony, and the maintenance of spontaneous beating, as compared to various controls including Matrigel, pure fibrin and collagen I as well as a fibrin-collagen I blend.

## 1. Introduction

Cardiovascular disease,, in particular, myocardial infarction, is the leading cause of mortality, with a toll of 31% of global deaths [1]. In myocardial infarction, cardiac ischemia leads to oxygen deficiency, cell injury and ultimately cell death [2]. Furthermore, the remaining heart tissues are subjected to a substantial structural and functional remodeling [3]. As adult cardiomyocytes possess only limited regeneration capability, the search for strategies to enhance heart tissue regeneration for a robust recovery after myocardial infarction became of paramount importance [4]. Our aim is to provide a simple, yet highly efficient in vitro model for cardiomyocyte differentiation and maintenance. The suggested model, based on a blend of decellularized cardiac extracellular matrix (dECM) and thrombin-coagulated fibrin, is designed to simultaneously recapitulate two main characteristics of a native cardiomyocyte environment [5]: extracellular matrix cues [6] and mechanical stiffness [7]. 

Our first cardiomyocyte environment element is the choice of a specific cardiac extracellular matrix. Hydrogels surfaced as promising candidates to simulate native tissue scaffolds due to their ability to support cell attachment, migration, proliferation and differentiation [8]. Furthermore, mixed cardiac cultures of both contractile and non-contractile cells [9] employing purified hydrogel extracts, such as collagen I, as well as mixtures of collagen I with Matrigel, were reported to promote spontaneous synchronized in vitro contractions [10,11,12]. In vivo, collagen foams and Matrigel were further shown to enhance the engraftment efficiency of the cardiac cells H9c2 [13]. 

More recently, integral decellularized myocardial extracellular matrix, dECM, has received increased attention. dECM was shown to potentiate cardiomyocyte differentiation and maturation in many cell lineages, including human embryonic stem cells and rat neonatal ventricular cardiomyocytes [6]. Porcine cardiac dECM hydrogels are particularly ultra-structurally similar to their human analogs [14] and, importantly, capable of providing functional benefits after myocardial infarctions in animal models [15]. Partial preservation of organ-specific cues [16,17,18], corroborated by growth factors contained in the dECM [6,19], is thought to be responsible for correct organ-specific cellular differentiation. Given that most studies focus on the development of a particular type of hydrogel, it is still unclear whether the additional complexity of dECM, as compared to purified extracts, benefits cardiomyocyte differentiation or electromechanical function, or whether the documented effects are rather mediated by a single major component such as collagen I. An important aim of this study is therefore to reveal potential advantages of dECM over the available commercial formulations such as collagen I or Matrigel.

Hydrogel extracts such as dECM, but also Matrigel or collagen, generally suffer from insufficient mechanical strength when compared to native cardiac muscle [6,19,20]. This is unfavorable because a substrate with a comparable stiffness to that of the native heart is known to be most conducive to the differentiation of heart cells into differentiated phenotypes [7]. The second goal of our in vitro cardiomyocyte niche model is therefore the achievement of appropriate mechanical stiffness. While synthetic tissue-engineered scaffolds made of polylactic acid and polyglycolic acid, polyurethane or poly(glycerol sebacate) provide mechanical strength, they are often affected by toxicity, a lack of compatible degradation pathways as well as by their immunogenic properties. In order to combine the favorable biological properties of hydrogels with adequate mechanical properties, various re-enforcement strategies have been tried. This includes covalent crosslinking [21] or combination with additional hydrogels [22]. For example, Williams and colleagues used enzymatic crosslinking with transglutaminase to strengthen a blend of fibrin and dECM from neonatal and adult rat hearts, followed by crosslinking with transglutaminase [23]. However, the lack of specificity of transglutaminase or chemical crosslinking led to concerns not only about cellular toxicity in vitro [23] but also about the possible side effects such as autoimmune reactions due to altered self-epitopes in vivo [24]. Therefore, we sought to explore here the use of fibrin obtained by the coagulation of fibrinogen with human thrombin [25] to improve both the stiffness in dECM-fibrin blends and their safety. Previous studies [23] reported that the use of fibrin, as opposed to synthetic polymers [22], is rationalized by the documented increase in efficiency of cardiac reprogramming [26]. 

To test the cardiogenic differentiation potential, H9c2 cell line was included as a model system [27]. Originally derived from rat embryonic ventricular heart tissue, H9c2 cells spontaneously differentiate towards skeletal muscle upon reaching confluence [27]. Furthermore, cardiac differentiation can be recovered upon exposure to retinoic acid [28] and further enhanced under a fibroblast-derived matrix [29]. Importantly, cell-friendly, rapid crosslinking with thrombin provides a simple opportunity to implement surface and 3D co-cultures with fibroblasts by seeding predefined co-culture mixtures. Our aim here is to assess the cardiogenic differentiation potential of thrombin-coagulated fibrin and porcine cardiac dECM hydrogels, as compared to the commercial controls Matrigel and collagen I. In addition, we sought to delineate the physiological electromechanical activity and synchronization of primary neonatal cardiomyocytes by addressing the development of the cardiac in vitro niche and its relevant cellular, mechanical and matrix components.

Hence, we proceeded to investigate synchrony, beating rate and recovery time of neonatal cardiomyocytes. Various control hydrogel compositions such as pure fibrin, collagen I and Matrigel were included to further substantiate the role of niche, the relevance of primary neonatal cardiomyocytes as well as the optimal definition of tissue engineering and transplantation matrices.

Figure 1 shows a brief flow diagram of the experimental settings and assays employed in this study, including extracellular matrix decellularization, hydrogel preparation based on dECM and characterization, and cardiac cell functionality in our genuine hydrogel. We envision several potential applications for our model such as drug screening and patch transplantation; it also simplifies the generation of cardiomyocytes and may provide building blocks for cell transplantation in regenerative medicine.

## 2. Results

### 2.1. Development and Characterization of the Hybrid Hydrogel 

#### 2.1.1. Decellularization Characteristics

We successfully decellularized porcine cardiac extracellular matrix. As shown in Figure 2A, the total amount of remaining DNA in the hydrogel after dissolving the lyophilized dECM powder in lysis buffer is less than 50 ng/mg of tissue, indicating an essentially complete removal of cells [30]. The concentration of the extracellular matrix components collagen and glycosaminoglycans (GAG) was measured in the native tissue and after decellularization and lyophilization, and it showed no loss of these components in the resulting dECM (Figure 2B,C). Hematoxylin–eosin (H&E) and specific staining for collagen (Sirius red) and elastin (Miller) confirmed the absence of cells or cell debris. At the same time, the presence of collagen and elastin in the matrix was documented after decellularization (Figure 2D). 

#### 2.1.2. Mechanical Properties

We next proceeded with the investigation of the relationship between the elastic modulus and the fibrinogen concentration in pure fibrin gels. These measurements will give us a good insight regarding the stiffness of the hydrogel. In line with previous reports [31], our analysis shows that elevated fibrinogen concentrations are associated with larger Young moduli, although at the highest concentrations, a saturation effect can be observed. 

The evaluation of the storage and loss moduli in dECM-fibrin gels in a linear temperature scan is presented in Figure 3B. At 37 °C, the Young’s modulus of the dECM-fibrin hydrogel stabilizes at about 21 kPa, falling into the reported range of native adult myocardium from 11.9 to 46.2 kPa [7]. This indicates that mechanical properties adequate for heart cell culture can be achieved via thrombin coagulation, a method that eliminates the involvement of crosslinking agents with low specificity [7,32]. The macroscopic examination of the gelated dECM-fibrin hydrogel revealed a firm, slightly elastic, and semi-transparent solid structure (Figure 3D).

#### 2.1.3. Gelation Time

Our measurements indicate that the gelation time of dECM in the absence of cells is about 10 min (Figure 3C). The addition of fibrinogen and thrombin to the hydrogel is associated with a reduction in gelation time of about 1–2 min. This gelation time is still sufficient for the manipulation of cells in 3D and is free of the risk of cell sedimentation. 

#### 2.1.4. Structural Characterization and dECM Stability

Figure 4A shows a scanning electron micrograph image of the dECM-fibrin hydrogel, while Figure 4B shows specifically the spatial distribution of rhodamine-labelled dECM within the dECM-fibrin composite. The electron micrography of dECM hydrogels revealed a heterogeneous structure of the hydrogel in both imaging modes (Figure 4A,B). This implies that coherent pieces of dECM subsist and are embedded into fibrin, rather than forming a spatially homogeneous mixture, with heterogeneity probably defining local niche structures. In other words, the rhodamine isothiocyanate stains the collagen strands, which is the main component of dECM. Consequently, Figure 4 shows that in a block of dECM-fibrin hydrogel, all points containing dECM are red in color. Based on Figure 4, there is a homogenously disrupted staining pattern throughout the block of hydrogel, which indicates that dECM is distributed uniformly throughout the mixture. The interrogation of dECM stable maintenance within the dECM-fibrin composite at 7 and 14 days revealed no loss of dECM (Figure 4C,D). The confocal stack of control dECM-fibrin hydrogel in the absence of rhodamine (Figure 4E) validates the specific detection of rhodamine-labelled dECM. As expected, virtually no autofluorescence was detected at the exposure settings employed in Figure 4B–D. 

### 2.2. Cell Seeding and Troponin T Expression on the Surface of the Hydrogel and 3D dECM-fibrin Hydrogel

To assess the capacity of various hydrogels to enhance cardiogenic differentiation in H9c2 cells, H9c2 cells and Nor-10 fibroblasts were seeded at a co-culture ratio of 30%:70% onto hydrogel slabs in 48 well plates. Appendix A confirms the impact of co-culturing with fibroblasts on H9c2 cells differentiation and the optimal percentage for the co-culture. Based on this data, the ratio of 30/70 of H9c2 cells to fibroblasts is critical for the differentiation. Troponin T expression on dECM-fibrin, Matrigel, collagen and on the control tissue culture plate was assayed after 1 week of differentiation (Figure 5A–D). Then, we proceeded with the quantification of the percentage of Troponin T expressed H9c2 cells co-cultured with fibroblasts on the different materials (Figure 5H). Our measurements indicate that the proportion of Troponin T expressed H9c2 cells is larger on the dECM-fibrin hydrogel than on cell culture plates (*p* = 0.0001) and Matrigel (*p* = 0.0024), but similar to that on collagen (*p* = 0.645). These results suggest that the collagen content in the dECM impacts on the expression of Troponin T of H9c2 cells. 

Having evaluated and, importantly, optimized the expression of Troponin T on the surface of various hydrogels, our next step was to proceed to the implementation in a genuine 3D co-culture configuration. In this experimental setting, H9c2 and Nor-10 cells were included in the dECM-fibrin hydrogel prior to gel formation. 

Figure 5E shows the Troponin T expression of H9c2 cells in the 3D hydrogel using Troponin T staining. The top and bottom view of the 3D hydrogel is shown in Figure 5F,G. Dashed white boxes in Figure 5F,G correspond to parts of the dECM-fibrin hydrogel that do not contain H9c2 cells, so no red staining is visible, confirming the absence of non-specific staining for Troponin T antibody. Dedifferentiation in 3D hydrogels was associated with cells that were attached and had spread and formed a network throughout the wells. Our measurements indicate that upon co-cultivation with fibroblasts 3D dECM-fibrin hydrogels, H9c2 cells expressed Troponin T at a rate of 83 ± 8%. H9c2 and fibroblast cells co-cultured in 3D showed significantly better results than 2D co-cultures on dECM-fibrin coated surfaces (*p* = 0.0007), indicating an advantage of the 3D configuration (Figure 5H). Co-culturing with Nor-10 fibroblasts, a 3D environment, and a collagen-based matrix showed a cardiac differentiation-promoting effect on H9c2 cells. However, it should be noted that a fundamental advantage of dECM over collagen could not be detected with H9c2. 

### 2.3. Neonatal Cardiac Cells on 2D and in 3D Hydrogels 

To substantiate the advantage of different hydrogels within a more physiologically relevant cell system, primary neonatal cardiomyocytes were cultured on surfaces (2D culture) of dECM-fibrin hydrogel. Various controls such as tissue culture plates, commercial matrices such as collagen I or Matrigel, as well as a fibrin-collagen I composite were involved. Fibrin-collagen I composite is the closest analog to dECM-fibrin with regard to its chemical composition, because around 70% of dECM is composed of fibrillar collagens, especially Collagen types I and V. With the help of Fluo-4 AM, time-lapse measurements of intracellular calcium ionic concentrations were recorded after 3 days of culture, in order to assess electrophysiological activity.

Representative videos are provided as Appendix A. Calcium oscillation video recordings served to evaluate local frequency (Figure 6A–E) and local phase (Figure 6F–J). The results essentially indicate an almost perfect synchrony of frequency (Figure 6E) and phase (Figure 6J) on the dECM-fibrin hydrogel. However, phase analysis of fibrin-collagen I revealed some individual desynchronized cells and regions with increased lag (Figure 6I). Matrigel, on the other hand, showed almost no phase synchronicity and low beating frequency compared to other experimental conditions (Figure 6G). This indicates that dECM-fibrin positively impacts the synchronization of calcium influx. 

Following the optimal synchronization of calcium influx features in the dECM-fibrin matrix in 2D, we investigated 3D cultures of neonatal rat cardiomyocytes in dECM-fibrin hydrogels compared to fibrin alone and fibrin-collagen I cultures.

The normal native beating rate of the neonatal rat heart is around 276–423 beats per minute (bpm) [33]. At the same time, the beating rate of isolated primary cells in vitro on standard cell culture conditions is about 100–115 bpm (manufacturer’s notice [34], confirmed in preliminary trials). The effect of the extracellular matrix environment on the beating rate is presented (Figure 6K). Pure fibrin gels sustain a relatively low beating rate at 77 bpm after 5 days of culture. At the same time, fibrin-collagen I (166 bpm, *p* = 0.0001 vs. fibrin) and dECM-fibrin cultures (206 bpm, *p* = 0.0001 vs. fibrin-collagen I) approach the expected beating rate of the neonatal heart. 

Throughout all conditions, the mechanical beating function gradually recovered, as indicated by a progressive increase in frequency (Figure 6K). This beating recovery started earlier in the dECM-fibrin hydrogel as compared to the other conditions.

By quantifying synchrony in neonatal cardiac cell cultures, we confirmed our findings (Figure 6L). Primary cardiomyocytes seeded in dECM-fibrin gel started contracting on the first day and continued for at least 10 days (Appendix A). In contrast, cardiomyocytes seeded in fibrin did not show synchronous beating until Day 5 and typically stopped beating after 7 days. Furthermore, cardiomyocytes cultured in fibrin-collagen I reached synchronicity at Day 3 but, similarly to fibrin, did not beat after 7 days. 

Results confirmed the superior performance obtained with dECM-fibrin hydrogel by analyzing the calcium imaging data (Figure 6A–J), making it the ideal hydrogel among the alternatives here for primary neonatal cardiomyocyte culture.

## 3. Discussion

Our study aimed to develop a hydrogel blend of fibrin and decellularized porcine cardiac matrix to improve the cellular niche for cardiomyocyte cell culture. For this, we successfully decellularized porcine ventricular tissue, with accurate maintenance of collagen, glycosaminoglycans and elastin components. The addition of fibrinogen to the newly designed dECM, followed by coagulation with thrombin, provided a Young modulus of about 20 kPa, which intimately reflects the range of adult cardiac tissue [7]. This system was employed in conjunction with fibroblast co-cultures to improve the differentiation capacity of the H9c2 myoblast cell line towards a cardiomyocyte phenotype (Appendix A). Finally, the results of the calcium imaging analysis in neonatal cardiomyocytes demonstrates the higher frequency, synchronicity, and earlier beating recovery in dECM-fibrin hydrogel.

An important task in the design of the hydrogel system was to attain the physical stiffness of a native myocardium (10–50 kPa range, [7,35]). To mechanically reinforce the intrinsically soft cardiac dECM, we chose the specific coagulation of fibrinogen by thrombin, as opposed to more generic agents such as transglutaminase [24,36] or genipin [30,37]. While thrombin-coagulated fibrin gels appear softer than similar gels crosslinked by transglutaminase [24], we were able to increase the concentration of fibrinogen to compensate for this limitation. As a result of using thrombin [38] for fibrin gel formation, we could avoid the cellular toxicity caused by transglutaminase crosslinking [24].

H9c2 cell line was used to investigate the cardiogenic differentiation. The combination of 3D embedding into dECM fibrin hydrogels with Nor-10 fibroblasts was already able to render the efficiency of Troponin T expression in the absence of exogeneous retinoids of over 80%. The advantage of adding fibroblasts or fibroblast-derived matrix in H9c2 differentiation has previously been demonstrated [29,39,40]. Compared to most previous studies, our set-up has a differentiation efficiency of about 85% [30,39]. Our stepwise optimization addressed three key elements of the cardiac cell environment: (i) choice of the extracellular matrix, (ii) ECM physical properties, and (iii) the co-culture composition [41]. We envisage that our strategy will simplify pharmacological screening, especially since retinoids, which are traditionally used for cardiogenic differentiation, broadly affect cellular processes beyond specific cardiac differentiation [42]. Furthermore, the system could be deployed for detailed investigations of the minute communication between cardiac fibroblasts and cardiomyocytes, which today remains only partially understood [43]. It is also important to note that the dECM-fibrin composite did not improve the efficiency of H9c2 Troponin T expression over the commercially available collagen I. Based on our observations, it appears that the expression of Troponin T in H9c2 cells is largely influenced by the collagen composition. This outlines the prime importance of collagen I for the cardiogenic differentiation in this model. While alone, these results do not provide minute supportive evidence, it is important to emphasize the advantageous trait of the rapid gelation afforded by fibrin, particularly in 3D embedding, as from a practical point of view, it greatly reduces cell sedimentation.

The ultimate goal of an in vitro cardiac model is to accurately replicate the electromechanical beating function. Our comparative culture of freshly isolated rat cardiomyocytes on dECM-fibrin, fibrin-collagen I and fibrin control hydrogels indicated that the dECM-fibrin composite significantly improved physiological beating frequency recovery and synchrony. In line with these results, dECM-fibrin also outperformed collagen I, Matrigel, tissue culture plates, and fibrin-collagen I regarding synchronization in calcium imaging. These differences could not be revealed in the H9C2 cell model.

As demonstrated by our experiments with these two cell groups, meaning H9c2 cell model and neonatal cardiac cell model, we believe that we can not necessarily draw the conclusion just by using H9c2. The main difference in maintaining the beating frequency and rapid recovery will emerge when the isolated neonatal cells are used.

Biochemical factors are crucial for the cardiogenic differentiation of H9c2 cells. The interaction with collagen I, probably through integrin binding [44,45,46,47], seems to be the key factor for the differentiation of H9c2 cells. Even the biochemical signals provided by Matrigel, which is composed primarily of collagen IV and laminin [48], considerably enhance H9c2 differentiation as compared to bare tissue culture plates.

Mechanical aspects should not be dismissed when characterizing primary cardiomyocytes. Optimal energy transmission to the substrate is expected to occur at near physiological stiffness (neither too soft, preventing force development, nor too stiff, preventing substrate deformation) [49,50,51]. Mechanical feedback [46] could therefore help in explaining why physiological frequency and improved synchrony for the neonatal cardiomyocytes are found for the composite hydrogels, but not for the softer collagen, Matrigel or pure fibrin or for the hard tissue culture plates.

Our investigations involving the primary cardiomyocyte system replicate important aspects of native cardiac physiology [52]. Here, a better geometrical connectivity due to improved cell spreading is expected to provide superior electrical synchronization, with reinforcement by mechano-electrical feedback on substrates with physiological stiffness [53]. Moreover, the primary cardiomyocyte population is expected to contain various pacemaker cells of different intrinsic frequencies [52]. An improved connectivity not only yields superior synchrony, but also a better spread of the highest pacemaker frequency.

Interestingly, this observation was not made in the differentiation of H9c2 cells, where collagen I and dECM-fibrin affected differentiation efficiency in the same way. In other words, primary cardiomyocytes are a better model for studying the effectiveness of hydrogels.

Altogether, our study indicates that the presence of collagen I along with its appropriate mechanical properties makes an important contribution to the several favorable properties of the dECM-fibrin composite. Indeed, despite its substantially lower collagen I concentration, the fibrin-collagen I composite nearly matches the performance of dECM-fibrin in terms of beating frequency [54]. Additional studies are required to further assess whether the remaining differences in synchronization efficiency, long-term maintenance of beating, and beating frequency are due to the quantitative differences in collagen I content or to the contribution of additional beneficial effects arising from non-collagen I components in the cardiac dECM. This question is of utmost importance in helping to create the rational design of drug screening and cell transplantation hydrogels that are supportive of cardiomyocyte differentiation and function. In our research, rapid synchronization and long-term electromechanical function are likely to be dependent on the precise composition of the cardiac dECM. However, commercial collagen I extracts are surprisingly effective at recapturing some of the dECM properties because of the presence of collagen in the dECM. dECM-fibrin contains various components in addition to collagen, such as elastin and GAGs, which sets it apart from a collagen hydrogel. Experiments using H9c2 cells could not show a significant distinction between our dECM-fibrin hydrogel and a collagen hydrogel. However, upon using neonatal cardiac cells and conducting further functional tests, a greater degree of variance was observed, highlighting the potential benefits of using dECM-fibrin hydrogel for cardiac cell culture.

In order to achieve rapid gelling and the physiological stiffness required to enhance physiological beating, thrombin-induced coagulation of fibrin is an effective, non-toxic pathway in combination with either dECM or pure collagen I.

## 4. Materials and Methods

### 4.1. Extracellular Matrix Decellularization and Characterization

Decellularized porcine cardiac extracellular matrix forms the basis of the dECM-fibrin hydrogel presented in this manuscript. The procedure for decellularization of the cardiac tissue builds on previously published work [55]. To summarize, porcine heart tissue was obtained from a local slaughterhouse and the ventricles cut into pieces of about 1 mm in thickness. The pieces were rinsed with deionized water and then stirred in 1% Sodium Dodecyl Sulfate (SDS) in a phosphate-buffered saline (PBS) solution for 48–72 h, at 4 °C, followed by 1% Triton X-100 for an additional 30 min. Finally, the preparate was stirred in deionized water overnight and freeze-dried. The resulting dECM powder was suspended in 0.1M HCl, followed by pepsin (Sigma P6887, Buchs, Switzerland) digestion for 72 h (100 mg of dECM and 10 mg of pepsin per 1 mL of HCl) [54]. The pH was then adjusted to 7.4 by gradually adding NaOH to yield a solution of 100 mg/mL dECM. dECM stock solution was finally obtained by adding Dulbecco’s Modified Eagle Medium DMEM (Thermofisher, Cat# 41965) to reach a final dECM concentration of 50 mg/mL.

To evaluate the extent of decellularization, the residual DNA content in both native and decellularized tissue was measured. For this, dECM (and in parallel, intact cardiac tissue) was dissolved in lysis buffer (0.5 M EDTA pH 8.0, 0.5% sodium dodecyl sulfate) and 100 µg/mL proteinase K (Sigma, P4850, Buchs, Switzerland) overnight at 55 °C [56]. The resulting suspension was vortexed, and the proteins precipitated with phenol-chloroform, followed by centrifugation at 13,700 rpm for 40 min. DNA was then isolated by recovery of the top (aqueous) phase, followed by addition of 0.5 mL ethanol, resuspension in deionized water, and quantification by Carry 50 spectrophotometer using a quartz cuvette at 260 nm. For further quantification of the composition of dECM, the collagen and Glycosaminoglycan (GAG) content in the dECM were measured using the Sircol™ Soluble Collagen Assay kit and Blyscan Sulfated Glycosaminoglycan Assay kit. Finally, histological sections of dECM were prepared by standard paraffin embedding. Sections were subjected to hematoxylin–eosin (H&E), Sirius red, and Miller staining before they were scanned by an Olympus VS120-L100 microscope slide scanner to analyze collagen and elastin expression.

### 4.2. Hydrogel Preparation

The preparation of dECM-fibrin hydrogel involves several steps. Firstly, pre-gel solution was prepared by mixing 100 µL of 50 mg/mL dECM stock solution, 528 µL of 50 mg/mL fibrinogen (Sigma, F3879, Buchs, Switzerland) solution, 100 µL of HEPES 1M pH 7.4 and 269 µL of DMEM. To induce gelling, 1.7 µL of Thrombin (Sigma, T1063, 250 U/mL, Buchs, Switzerland) and 1.3 µL of calcium chloride (CaCl_2_) 1M were added to the pre-gel solution before incubation, at 37 °C. The final composite gel featured 5 mg/mL dECM and 26.4 mg/mL fibrinogen.

Matrigel (Sigma, E1270, solution supplied at 9 mg/mL, Buchs, Switzerland) was diluted to 3 mg/mL with DMEM before gelation. Collagen I (Sigma, C4243, solution supplied at 3 mg/mL, Buchs, Switzerland) was diluted to 1 mg/mL with DMEM prior to gelation. This also neutralized the pH. The final fibrin-collagen I composite was obtained by mixing 157 µL of 3 mg/mL collagen stock solution, 314 µL of DMEM, followed by 528 µL of 50 mg/mL fibrinogen before the induction of gelling upon addition of 1.3 µL of CaCl_2_ and 1.7 µL of thrombin (250 U/mL).

### 4.3. Mechanical Properties

To measure the mechanical properties, fibrin and dECM-fibrin hydrogels were prepared and pipetted into cryovials and incubated for 30 min, at 37 °C, to gel. Upon removal from the cryovials, samples were subjected to compression testing using a dynamic mechanical analyzer (TA Instruments DMA Q800). The storage and loss moduli are defined as follows [57]:(1)$$\mathrm{Storage}:\;E^{\prime }=\frac{\sigma_{0}}{\varepsilon_{0}}\;\cos\delta ,\;\;\;\;\mathrm{Loss}:\;E^{\prime \prime }=\;\frac{\sigma_{0}}{\varepsilon_{0}}\;\sin\delta$$
where σ_0_ is stress, ε_0_ is strain, and δ is the phase angle or phase lag between the stress and strain.
(2)$$\mathrm{The}\;\mathrm{Young}’s\;\mathrm{modulus}\;\mathrm{is}:\;\;\;\;\;\;\;E=\sqrt{{E^{\prime }}^{2}+{E^{\prime \prime }}^{2}}$$

### 4.4. Microstructure Characterization

The microstructure was analyzed by scanning electron microscopy (SEM). For this, the hydrogel sample was frozen and lyophilized. The lyophilized hydrogel sample was coated with a few nanometers (5–10 nm) of gold prior to performing SEM imaging.

### 4.5. dECM Stability Study with Fluorescently Labeled dECM

To investigate the stability of the dECM in our composite hydrogel, we fluorescently labeled the dECM using rhodamine isothiocyanate [53]. This is important for attaining a homogenous hydrogel throughout the sample. To achieve this, 300 mg dECM powder was suspended in 10 mL of 0.1M HCl and the pH adjusted to 10.3 with 0.9 mL NaOH 1M and 0.3 mL of Na_2_CO_3_ 1M. Then, 6 mg of rhodamine isothiocyanate was dissolved in 10 mL of isopropanol and then added to the reaction mix. After overnight incubation, at 37 °C, the dECM was precipitated and repeatedly washed with excess isopropanol, until a clear supernatant and strongly stained precipitate were obtained. The precipitate was air-dried overnight. dECM-fibrin hydrogels were prepared with fluorescently labeled dECM, using identical procedures as for unlabeled dECM. Fluorescent imaging was conducted after 1, 7, and 14 days of incubation in PBS, at 37 °C.

### 4.6. In Vitro Cell Studies

#### 4.6.1. H9c2 Cell Culture

H9c2 cells were obtained from the European Collection of Authenticated Cell Cultures (ECACC) (Lot# 17A028). Cells were cultured in DMEM medium supplemented with 10% fetal bovine serum and 1% penicillin and streptomycin in 75 cm^2^ tissue culture flasks, at 37 °C, and 5% CO_2_ in a humidified atmosphere. To avoid loss of differentiation potential, H9c2 cells were passaged before reaching 70–80% confluency according to the supplier recommendations [29].

#### 4.6.2. Nor-10 Cell Culture

Nor-10 (ECACC 90112701) cells were obtained from the European Collection of Authenticated Cell Cultures. Cells were cultured in DMEM medium supplemented with 10% fetal bovine serum and 1% penicillin and streptomycin in 75 cm^2^ tissue culture flasks, at 37 °C, and 5% CO_2_ in a humidified atmosphere. They were split before reaching 70–80% confluency according to the supplier’s notice [58].

#### 4.6.3. H9c2 Differentiation

To study the capacity of the dECM-fibrin composite and various control hydrogels intended to enhance the cardiogenic differentiation of H9c2 cells, we modified an established H9c2 differentiation protocol [28]. Differentiation was initiated by lowering serum content to 1% for 5 days. The final one to two days of the culture process occurred in expansion medium (DMEM with 10% FBS), as described in [28]. The original protocol also uses retinoic acid [28]. However, preliminary trials showed that co-culturing with Nor-10 fibroblast could partially replace the addition of retinoic acid on tissue culture plates (Appendix A). Therefore, to test the cardiogenic differentiation potential of various hydrogels, we used co-cultures with a seeding ratio of 30% H9c2 and 70% Nor-10 fibroblasts, inspired by the native fraction of cardiomyocytes in the heart [8,40], without addition of exogenous retinoids. This modification permits us to assess the effect of the hydrogels in the absence of strong chemical induction and also avoids problems with retinoic-acid-induced fibrinolysis [59].

#### 4.6.4. Surface Cell Seeding on Hydrogel

To study the biochemical influence of different hydrogels on cardiomyocyte differentiation in a monolayer geometry, we prepared ca. 0.5 mm high hydrogel blocks in a 48-well plate. For this, a mix of 50 µL of dECM-fibrin, collagen I or Matrigel was gelled in those wells that were of interest. H9c2 cells and Nor-10 fibroblasts at a ratio of 30%:70% and density of 2.5 × 10^5^ cells per cm^2^ were applied on top of hydrogels. The differentiation protocol was conducted as described above: 5 days with 1% FBS in DMEM, followed by 1 day of 10% FBS in DMEM.

#### 4.6.5. Cell Seeding in 3D Hydrogel

To assess whether differentiation could be improved in 3D vs. 2D, we suspended a pellet of Nor-10 and H9c2 cells (70%:30% ratio) in various pre-gel solutions at 10^6^ cells/mL. Thrombin and calcium chloride were also added to the solution in the case of fibrinogen-based gels. Next, 200 µL pre-gel cell solution was rapidly pipetted into a 48-well plate. The hydrogel was allowed to solidify in the incubator upon thrombin’s reaction in the presence of fibrinogen or by pure self-assembly for the collagen I and Matrigel controls. The differentiation protocol with transient serum reduction to 1% was conducted as described above.

#### 4.6.6. Immunostaining and 3D Imaging

Cell culture samples were fixed in 4% paraformaldehyde (PFA) for 20 min, at room temperature (RT), before addition of 0.1% TritonX-100 to allow permeabilization of the cells for 30 min. Phalloidin-Atto 488 (Sigma 49409, Buchs, Switzerland) (1:50) was employed for 45 min, at 4 °C, to visualize the actin filaments [60].

Differentiation was assessed by measuring the fraction of H9c2 cells positive for troponin T (Sigma, #T6277, Buchs, Switzerland) by immunofluorescence. As this antibody detects both cardiac and skeletal muscle troponin T, we also evaluated the cardiac differentiation on dECM-fibrin samples using an antibody specific to cardiac troponin T (Abcam, #ab8295). This data is provided in the Appendix A.

For immunofluorescence, the cells were first blocked with 1% BSA for 1 h, at 37 °C. Troponin T primary antibody (Sigma, T6277, 1:50, Buchs, Switzerland) was then added, followed by incubation overnight, at 4 °C. Alexa-568 secondary antibody (Sigma, A10037, 1:1000, Buchs, Switzerland) was added after washing with DPBS (Gibco 2062235) and incubated for 45 min, at 37 °C, prior to washing with DPBS and staining with 4′,6-diamidino-2-phenylindole (DAPI, 1:2000 from 5 mg/mL stock solution) for 5 min. DAPI was replaced with DPBS before imaging the cells under a fluorescence microscope. The differentiation magnitude was quantified as the surface area covered by cells expressing troponin T (%) relative to the total area of the fluorescent images and normalized to the percentage of H9c2 cells seeded in the co-culture. It should be noted that this approach assumes a certain basal cell growth in the experimental conditions involving low serum content.

#### 4.6.7. Neonatal Cardiomyocytes Isolation

To study the effect of the thrombin-coagulated dECM-fibrin hydrogel on primary cardiomyocytes, we isolated primary rat neonatal cardiomyocytes using the Pierce Primary Cardiomyocyte Isolation Kit according to the manufacturer’s instructions [61]. Organ harvesting was performed on sacrificed control animals from unrelated experiments according to license VD 3290 designed for this specific purpose.

#### 4.6.8. Calcium Imaging

Primary rat cardiomyocytes (2.5 × 10^5^ cells/cm^2^) were seeded onto hydrogel slabs in 48-well plates (50 µL of hydrogel per well). Tissue culture controls were prepared by skipping the hydrogel polymerization step. After 3 days of culture in DMEM with 10% FBS, the cultures were loaded with 2 μM Fluo-4 AM (F14217) for 30 min, at 37 °C. Calcium transients were then recorded using fluorescent microscopy, at room temperature. Temporal peak detection was based on a custom ImageJ plugin, implementing the publicly available ‘Find peaks’ function of Octave [62] in Java. The plugin that we developed for this analysis is available as Appendix A, but also for downloading at https://github.com/tbgitoo/calciumImaging, along with the source code and a user manual. We used this plugin to evaluate local beating frequency and temporal phase shift from the calcium imaging videos.

#### 4.6.9. Beating Characteristics

To assess the contractile properties of primary cardiomyocytes in 3D cultures, primary rat cardiomyocytes were suspended at a cell density of 10^6^ cells/mL in pre-gel mixtures of dECM-fibrin, fibrin, and fibrin-collagen I composite. Gelling of 200 μL of cell-hydrogel mixture per well in a 48-well plate was followed by visualization and time-lapse monitorization of cultures. Synchrony and onset of beating was judged visually on a per-well basis. To quantify the mechanical beating rate of the cells, movies were acquired through a video camera-equipped microscope, while maintaining the samples, at 37 °C, in a temperature-controlled chamber. For quantification of the beating frequency, we used the Pulse software (Cellogy Inc., Menlo Park, CA, USA) [63].

### 4.7. Statistical Analysis

Data were compared using unpaired t-test (two-tailed, equal variances) in the GraphPad software. Error bars represent the mean ± standard deviation (SD) of the measurements (N.S. *p* > 0.05, * *p* < 0.05, ** *p* < 0.01, and *** *p* < 0.001).

## 5. Conclusions

In this study, we investigated thrombin-coagulated fibrin, admixed with dECM from porcine ventricular heart tissue, not only for its propensity to induce cardiac differentiation in the H9c2 cells but also for its capacity to maintain physiological, synchronous beating in primary cardiomyocytes. We found that mechanical properties can be adjusted to match those of the native heart tissue. Importantly, the choice of thrombin as a specific crosslinking agent eliminates concerns of toxicity of the generic agents. We successfully tested the system in the context of highly efficient cardiac differentiation of the myoblast cell line H9c2. We demonstrated highly efficient recovery, synchrony and near physiological beating frequency in neonatal cardiomyocytes. The combination of collagen I content and appropriate mechanical stiffness explains most of the favorable properties, although further investigations are needed to assess whether the non-collagen I components of dECM may be responsible for subtle differences in the early synchronization and long-term maintenance of primary cardiomyocytes.

We envisage the future use of this system both as a robust and economical cardiac 3D cell model for drug screening and as a building block for 3D printing, tissue engineering, and transplantation in regenerative medicine.

## Figures and Tables

**Figure 1 ijms-24-02842-f001:**
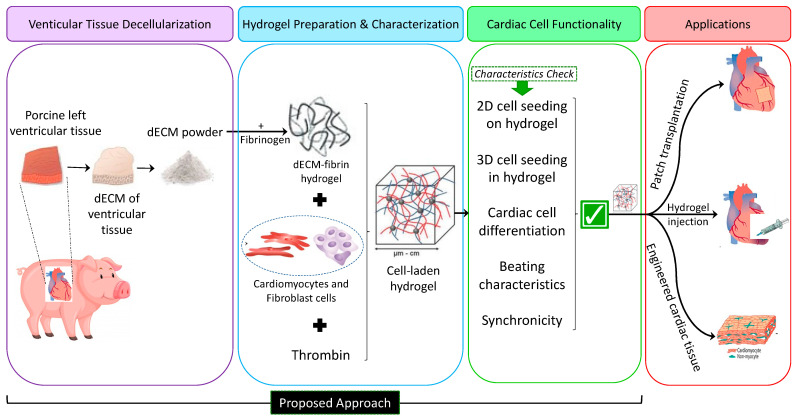
Flow diagram depicting the experimental approach in this study. Phase 1: dECM-fibrin hydrogel preparation, Phase 2: Characterization of the hydrogel. Phase 3: Investigating the functionality of cardiac cells in the hydrogel. The present study focuses on Phases 1 and 2.

**Figure 2 ijms-24-02842-f002:**
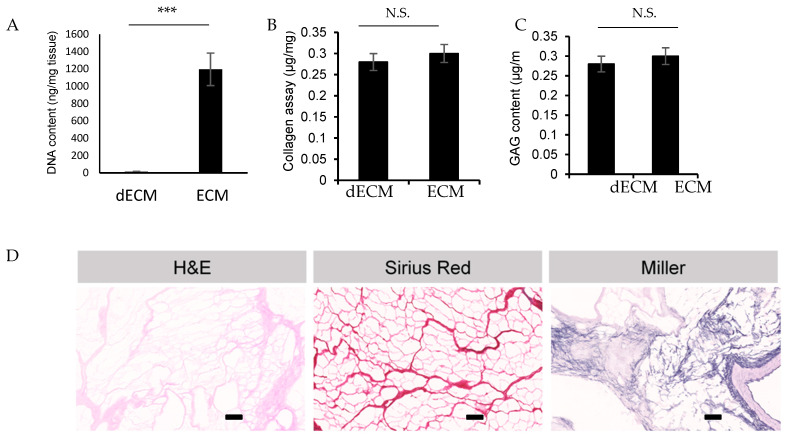
dECM characterization. Content of DNA (**A**), Collagen (**B**), and GAG (**C**) in the dECM after lyophilization and dissolving in lysis buffer is presented. All amounts relative to the weight of fresh tissue used. (**D**) H&E, Sirius red and Miller staining show nuclei, collagen and elastin, respectively, to prove removal of the DNA and the presence of collagen and elastin. *** = significant difference for *p* < 0.001 (N = 3–4 per condition), N.S. = Not Significant for *p* > 0.05. Scale bar: 100 µm.

**Figure 3 ijms-24-02842-f003:**
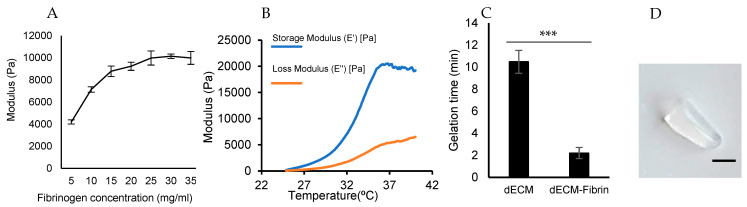
Mechanical characterization. (**A**) The stiffness of fibrin gels at different concentrations at room temperature is shown. This stiffness allows an accurate determination of the optimum concentration of fibrinogen in the hydrogel. (**B**) Storage and loss moduli of the dECM-fibrin hydrogel in a temperature scan (from 25 to 40 °C) is presented. The calculation of hydrogel’s Young’s modulus revealed comparable values to the native heart. (**C**) Depiction of the gelation time of the dECM and dECM-fibrin hydrogels, at 37 °C. Our experiments demonstrate that 2 min gelation are sufficient for secure handling/manipulation of dECM-fibrin hydrogels. *** = significant difference, *p* < 0.001 (N = 3–4 per condition). (**D**) Macroscopic appearance of the gelated hydrogels (Scale bar: 5 mm).

**Figure 4 ijms-24-02842-f004:**
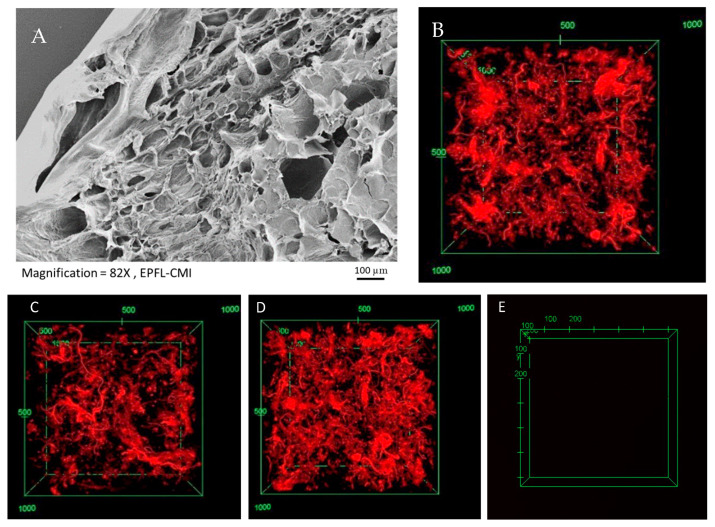
Microstructural analysis of dECM-fibrin gels. (**A**) Scanning electron microscopy image of the dECM-fibrin hydrogel is depicted. (**B**–**D**) dECM was labeled with rhodamine isothiocyanate before being subjected to gelation with fibrin. Confocal images after (**B**) 1 day, (**C**) 7 days, and (**D**) 14 days are pictured. (**E**) dECM-fibrin hydrogel without rhodamine labelling is shown. The distribution of dECM is locally heterogeneous. No significant loss of dECM occurs during this time. Scale bar: 100 µm. The hydrogel boxes in (**B**–**D**) are 1000 µm by 1000 µm.

**Figure 5 ijms-24-02842-f005:**
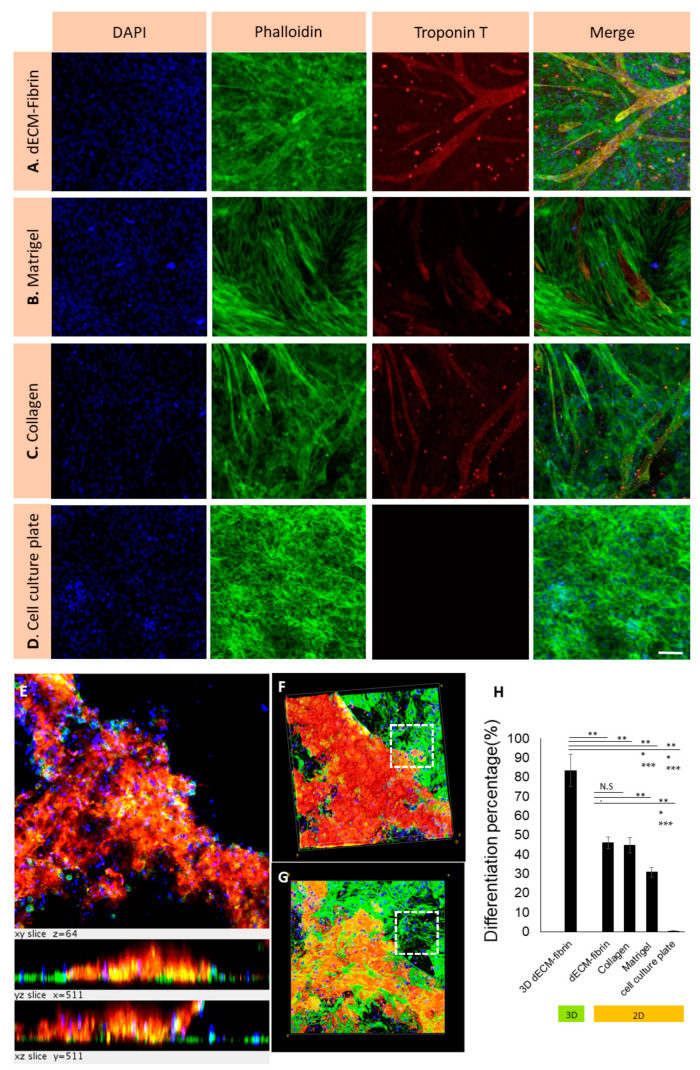
Expression of Troponin T in H9c2 cells in a co-culture with fibroblasts on different hydrogels (2D) and in the dECM-fibrin hydrogel (3D) in the absence of retinoic acid. Expression of Troponin T in dECM-fibrin (**A**), Matrigel (**B**), collagen I (**C**), and cell culture plates (**D**) after 7 days of differentiation is shown. Confocal images of H9c2 cell differentiation in a 3D co-culture featuring H9c2: fibroblasts in a 30%:70% ratio in dECM-fibrin hydrogel after 7 days is presented (**E**). Cells in dECM-fibrin hydrogel are shown: top view (**F**) and bottom view (**G**). Dashed white boxes in (**F**,**G**) confirm the absence of non-specific staining for Troponin T antibody (**H**) Percentage of Troponin T expressed H9c2 cells on different hydrogels in 2D and 3D is presented. Troponin T-specific staining (red) shows the differentiated cells. Phalloidin-specific staining (green) indicate the actin filaments of all cells. DAPI (blue) stains the nuclei. Scale bar: 100 µm. The hydrogel boxes in (**E**–**G**) are 511 µm by 511 µm. (*p* = 0.0024 for dECM-fibrin vs. Matrigel, *p* = 0.0001 for dECM-fibrin vs. cell culture plate, and *p* = 0.645 for dECM-fibrin vs. collagen.) N.S. = *p* > 0.05, * = *p* ≤ 0.05, ** = *p* ≤ 0.01, *** = *p* ≤ 0.001.

**Figure 6 ijms-24-02842-f006:**
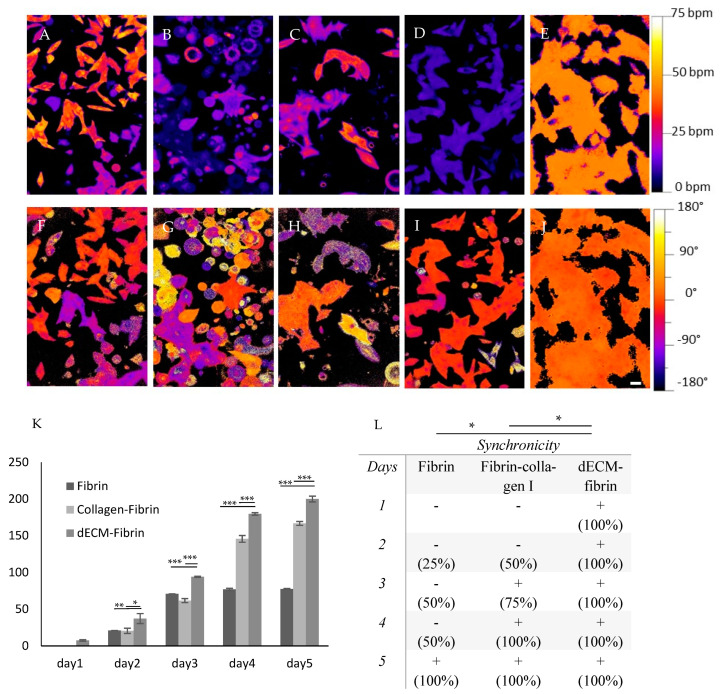
Calcium transients and beating characteristics of cardiomyocytes interacting with different hydrogels. (**A**–**J**) Calcium imaging for neonatal cardiomyocytes seeded onto different hydrogels (2D culture) and tissue culture plates (control). (**A**–**E**) Frequency (**A**–**E**) and Phase (**F–J**) are presented. Positive values indicate earlier beating, while negative values indicate retardation. Hydrogels used in calcium imaging: tissue culture control (**A**,**F**), Matrigel (**B**,**G**), collagen I (**C**,**H**), fibrin-collagen I (**D**,**I**) and dECM-fibrin (**E**,**J**). (**K**) The beating rate for the first 5 days for neonatal cardiomyocytes seeded 3D in fibrin, fibrin-collagen I and dECM-fibrin hydrogels is shown. (**L**) Synchronization as the percentage of synchronously beating wells (four wells per condition) over a period of 5 days in the 3D hydrogels is presented. * = *p* ≤ 0.05, ** = *p* ≤ 0.01, *** = *p* ≤ 0.001. Scale bar: 100 µm.

## Data Availability

Calcium imaging plugin can be found in the following link: https://doi.org/10.5281/zenodo.7575764.

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
