# Peer review of "Highly Efficient Cardiac Differentiation and Maintenance by Thrombin-Coagulated Fibrin Hydrogels Enriched with Decellularized Porcine Heart Extracellular Matrix"

_ijms, 2023, doi:10.3390/ijms24032842_

Round 1

Reviewer 1 Report

Authors presented a research article focused on the fabrication of thrombin-coagulated fibrin hydrogels enriched with decellularized porcine heart extracellular matrix and its improved function to differentiate cardiac cells. I do believe that the work is well-structured and would be of interest for a broad audience.

Few comments should be addressed by authors.

1.- Graphical abstract blurry

2.- Why authors don't include the non-loaded hydrogel in the mechanical assays (fig3a).

3.- This paper is already published online in biorxiv.

Author's should cite this paper? Include them?

Author Response

Dear reviewer, 

Thank you very much and Best regards,

Author

Reviewer 2 Report

Authors present interesting findings on the effects of a newly developed extracellular matrix on the ability of H9c2 cells to differentiate spontaneously. The authors have a well-organized manuscript with a well-conducted study. I request a few clarifications so that I can be more convinced of the data.

Major comment:

Can authors confirm that the ECM does not have non-specific staining for troponin antibody? Control images are needed, as DAPI and troponin are not overlayed well, and it appears that the matrix is stained rather than individual cells. Additional 20x magnification may also be helpful in confirming troponin staining of individual cells.

Minor comment:

Magnification should be indicated in legends of figures for which images are displayed.

I do not see supplementary materials attached.

Author Response

Dear Reviewer, 

Best regards,

Authors

Reviewer 3 Report

Navaee et al. performed an in vitro experimental study to establish an efficient in-vitro 3D cardiac cell culture model by thrombin-coagulated fibrin hydrogels enriched with decellularized porcine heart extracellular matrix. The results revealed the enhanced cardiogenic differentiation in the H9c2 myoblast cells co-cultured with Nor-10 fibroblasts and the enhancement of differentiation efficiency by 3D embedding of rat neonatal cardiomyocytes. The results seem to be interesting, however, the study looks stayed too preliminary. The conclusive remarks are not likely to be strongly supported by scientific evidences in the manuscript.

1.  In the Abstract, authors proposed that collagen content in the dECM is the key factor required for efficient differentiation of H9c2 cells and that use of dECM-fibrin was associated with substantial advantages for the cultivation of neonatal cardiomyocytes. However, there is no clear evidence to support this conclusion:

1) there is no experimental group setting to show that collagen content in the dECM is the key factor required for efficient differentiation of H9c2 cells. With the results indicating similar differentiation efficiency of H9c2 cells between Collagen group and dECM group compared to Matrigel group, it is hard to conclude that collagen content in the dECM is the key factor required for efficient differentiation of H9c2 cells.

2) In Fig.5,  with the levels of Troponin T in different groups, we cannot judge cardiogenic differentiation in the H9c2 myoblast cells. To prove cardiogenic differentiation in the H9c2 myoblast cells, a set of cardiogenic markers should be analyzed. Also it is not clear whether co-cultured Nor-10 fibroblasts might play a critical role for the potential cardiogenic differentiation (there is no control group for it in the manuscript.).

3) In Fig.6, it is not clear what is advantage of a dECM cell culture model over conventional collagen-Fibrin group.

2. The figures should be properly re-arranged.

Fig.2 : Fig2A is missing.

Fig.4: The figure contains very limited information as microstructural analysis of dECM-fibrin gels.

Fig.5: There is no description of Fig.5E in the figure legend. In Fig.5H, the number of Asterix is not matched with the number of Bars.

Supplementary data were not provided, although authors mentioned the results in the result sections (Page 8).

Fig.6: Fig6A, F and L are missing. Statistical remarks are wrongly located.

3. In each results section, for me, it is very hard to find the major findings and messages easily, probably because  the text arrangements do not seem to be properly organized.

4. In my opinion, the following descriptions or conclusive remarks in the Discussion section are not fully supported by the experimental proofs:

 1) Line 309-311: “This system was employed in conjunction with fibroblast co-cultures to improve the differentiation capacity of H9c2 myoblast cell line toward a cardiomyocyte phenotype.”

2) Line 311-313: ” Finally, our experimental setting reveals ensures the maintenance of electromechanical activity in neonatal cardiomyocytes as demonstrated by quantifications of beating activity as well as calcium imaging.”

3) Line 320-323: ” we could compensate for this caveat by increasing the fibrinogen concentration.. The specificity of thrombin[51] is advantageous for both in-vitro and anticipated in-vivo applications, and allowed us to avoid cellular toxicity triggered by transglutaminase crosslinking[24]”

4) Line 328-329: ”our experimental design achieved a higher efficiency in differentiation when compared to previous reports[52][30].”

5) Line 336-337: ”It is also important to note that no specific advantage of the dECM-fibrin composite over commercially available collagen I could be documented in regard to H9c2 differentiation efficiency.”

Versus.

Line 344-345: “specific advantages of the dECM-fibrin composite as this was associated with a substantially improved recovery of physiological beating frequency and reacquisition of synchrony.”

: There is no discussion about this discrepancy between these different results.

6) Line 349-351: “Our experiments show that the requirements for maintenance of rapid, synchronous beating in primary cardiomyocytes only partially overlap with the requirements for successful cardiogenic differentiation of the H9c2 cells.”

: I do not understand what the statement stands for.

7) Line 352-354: “The interaction with collagen I, probably through integrin binding[56][57][58][59], seems to be the key factor, nearly regardless of the mechanical properties of substrates.”

8) Line 357-359: “Cell adhesion is associated with higher frequency and improved synchrony in primary cardiomyocyte cultures as evidenced by cell spreading experiments presented in Figure 6.”

9) Line 375-378: “This subtly graded response to both biochemical and mechanical substrate properties contrasts with the differentiation of H9c2 cells, where presence of collagen I was as efficient as the dECM-fibrin composite in inducing cardiogenic differentiation, in essence independently on mechanical properties.”

10) Line 389-391: “…..long-term maintenance of electromechanical function might be dependent on the subtleties of the complex composition of cardiac dECM,….”

11) Line 393-395: “…thrombin-induced coagulation of fibrin is a successful, non-toxic pathway to achieve rapid gelling and physiological stiffness enhancing physiological beating.”

5. Many of spacing errors, typos and labeling mistakes in Figures are found throughout the manuscript. I think it would be better if co-authors should have participated in finalizing the manuscript before submission.

Overall, as the reviewer´s point of view, in the present study, authors seem to try to set too many goals hardly to achieve within a scope of a single manuscript. In contrast to the goals and conclusive remarks presented in the manuscript, the presented data looks still preliminary and seems to require more additional supporting evidences, reaching to the conclusions that authors have proposed in the Abstract and Discussion section.

Therefore, I believe that the manuscript should be re-considered following large-scale editing and clearer setting of aims for study with additional supporting data.

Author Response

Dear Reviewer, 

Best regards,

Authors

Round 2

Reviewer 1 Report

ok

Author Response

Dear Reviewer, 

We really appreciate your time and kind consideration. 

Best regards,

Authors

Reviewer 2 Report

Authors have adequately addressed comments.

Author Response

(The authors gave the same response as above.)

Reviewer 3 Report

Please find enclosed attached file.

Author Response

Dear Reviewer, 

Thank you very much for your time and for providing great feedback on our manuscript. Please find our response to your comments in the attached file. 

Best regards,

Authors 

Round 3

Reviewer 3 Report

I believe that the revised manuscript has been sufficiently improved for publication in IJMS.